# Effect of Remineralizing Agents on Shear Bond Strength of Orthodontic Brackets—In Vitro Study

**DOI:** 10.3390/children10020268

**Published:** 2023-01-31

**Authors:** Alexandrina Muntean, Cristina-Maria Dârgău, Mariana Pacurar, Simina Neagoe, Ada Gabriela Delean

**Affiliations:** 1Department of Paediatric Dentistry, Faculty of Dental Medicine, Iuliu Hațieganu University of Medicine and Pharmacy, 400083 Cluj-Napoca, Romania; 2Faculty of Dental Medicine, Iuliu Hațieganu University of Medicine and Pharmacy, 400083 Cluj-Napoca, Romania; 3Department of Orthodontics, Faculty of Dental Medicine, University of Medicine, Pharmacy Science and Technology G. E. Palade, 540142 Targu Mures, Romania; 4Department of Periodontology, Faculty of Dental Medicine, University Titu Maiorescu, 040441 Bucharest, Romania; 5Department of Conservative Dentistry, Faculty of Dental Medicine, Iuliu Hațieganu University of Medicine and Pharmacy, 400083 Cluj-Napoca, Romania

**Keywords:** shear bond strength, maximum load, fixed orthodontic treatment, bracket, adhesive, remineralizing agent

## Abstract

Orthodontic treatment can be effective only with the proper adhesion strength of the bonded elements on the teeth. The aim of the study was to analyze the influence of different remineralization products on the brackets (Evolve Low Profile Brackets 0.022 Roth prescription (DB Orthodontics Ltd., Silsden, England) shear bond strength (SBS)). In all, 40 teeth were investigated for this study; n = 30 demineralized (immersed in 0.1% citric acid for 30 min, twice a day, for 20 consecutive days) and n = 10 immersed only in artificial saliva. After the demineralization process, remineralization agents were applied to each group (n = 10): Group I: Elmex Sensitive professional^®^ toothpaste (CP, Gaba GmbH, Witten, Germany) and GC MI Paste Plus^®^ (GC, Leuven, Belgium), Group II: Elmex Sensitive professional^®^ toothpaste (CP, Gaba GmbH, Germany) and GC Tooth Mousse^®^ (Leuven, Belgium), Group III: Elmex Sensitive professional^®^ toothpaste (CP, Gaba GmbH, Germany). For the teeth in control group C, Elmex Sensitive professional^®^ toothpaste was used. SBS tests were performed by means of an advanced materials-testing machine that generated maximum load and tensile strength values. The data obtained underwent statistical analysis (ANOVA and Tuckey test) with a statistical threshold of *p* < 0.05. The SBS values were higher for group II (14.20 MPa) and I (10.36 MPa) and lower for group III (4.25 MPa) and C (4.11 MPa), with statistically significant differences between groups I and II when compared with groups III and C (*p* < 0.05). In conclusion, GC Tooth Mousse^®^ and MI-Paste Plus^®^ have no adverse effect on brackets SBS and are recommended to be used for enamel remineralization during orthodontic treatment.

## 1. Introduction

The tooth is a complex structure, which includes both hard, mineralized tissues (enamel and dentin) and soft tissues (dental pulp). The peculiarity of dental enamel is that it does not regenerate ad integrum, from a biological point of view but can benefit from mineral loss compensation through remineralization processes. For this reason, the preservation of the tooth enamel structure is a specific requirement for preventive dentistry, given that this method of regenerating dental structures has shown encouraging results, but there are still many steps to go until the complete restoration of this tissue [1,2].

Demineralization of dental enamel is the process by which mineral ions are removed from the hydroxyapatite crystals found in this hard dental tissue. The process takes place on the surface of the tooth, as a result of the pH variations in the oral cavity, at the interface with the mucobacterial plaque. It is possible that a significant quantity of ions can be removed from the hydroxyapatite crystals to a depth between 1 and 10 microns without affecting the integrity of the tooth; however, there is a risk of hypersensitivity to external factors such as thermal (hot, cold) or mechanical (pressure). The demineralization process is reversible only if the oral ecosystem favors mineral intake, with the restoration of the hydroxyapatite crystal structure [3,4].

The histochemical changes appearing in the initial stages of tooth enamel demineralization have been the subject of many studies with the specific purpose of identifying the reversible stages [1,5].

Saliva represents the oral fluid with the effect of neutralizing and removing acids, as well as the main source of inorganic ions that can initiate remineralization processes, since dental enamel does not contain cellular elements and has no repair capacity. In addition to its cleaning and antibacterial properties, saliva acts as a constant source of calcium and phosphate that helps to maintain a supersaturation in minerals. It inhibits demineralization during periods of low pH and ensures remineralization of the tooth when the pH returns to neutral values. Further, the pH increases rapidly to values above neutral when saliva secretion is stimulated [6,7].

Fixed orthodontic appliances are most often used for the treatment of dental and occlusal anomalies. It is suggested that a minimum shear bonding strength (SBS) of 5.9–8 MPa between orthodontic brackets and teeth would be adequate for clinical orthodontic tooth movement. The shear bond strength of the brackets depends on many factors, such as the structure of the tooth, the etching time and the acid used, the type of bracket and the orthodontic technique used, or even previously done procedures on the teeth, such as teeth whitening with 35% hydrogen peroxide [8].

Numerous strategies have been used to avoid demineralization in tooth surfaces when using fixed orthodontic appliances, one of which is the topical application of fluoride. For individuals receiving orthodontic treatment, fluoride can be given in a variety of ways, including toothpaste, mouthwash, gels, varnishes, and fluoride-releasing bonding materials. Fluoride has been shown to encourage remineralization across orthodontic brackets at low doses. Adhesives containing fluoride may release and absorb fluoride to shield teeth from erosion. Only a minor portion of the fluoride is still released over the long term because most of it is released during the setting reaction. Increased fluoride release while preserving bond strength has been achieved by combining glass ionomer cements (GICs) and composite resins. A novel strategy has been devised to shield the enamel from tooth erosion: resin infiltration. To penetrate and guard against developing erosive lesions, it has been advised to utilize enamel-penetrating, low-viscosity light-curing resins. This method places a diffusion barrier between the enamel and the acidic attack in order to occlude the pores of untreated enamel lesions. An important issue in orthodontics has been bonding orthodontic materials to a compromised tooth structure. Therefore, the usage of strong adhesives is required. In their study, Nebras et al. concluded that pretreatment with fluoride agents led to a decrease in the SBS of the brackets. Fluoride-releasing agents conducted to an enhanced bonding to eroded enamel [9].

Studies have shown that a demineralized or fluorosis-damaged enamel surface significantly reduces the adhesion of the brackets. The research also indicates that fluoride agents reduce SBS at the enamel–bracket interface, but the data on the effect of remineralizing agents on mechanical properties are controversial [10].

Casein-phosphopeptide-amorphous calcium-phosphate (CPP-ACP), a milk protein derivative, has also been suggested for enamel remineralization and caries prevention. Through the localization of the ACP on the tooth surface, the application of CPP-ACP results in a saturation of the enamel structure [10,11].

Short-term clinical investigations and several caries models have shown the potential anticariogenicity of CPP-ACP. The substance seems to be extremely safe for usage in foods, dental professional items, and oral care products for humans. Recaldent^®^, an alkaline, stable, and highly soluble substance, is currently present in sugar-free gum and mints as well as dental professional goods (Tooth Mousse^®^). In fact, numerous randomized, double-blind, in situ clinical investigations have shown that CPP-ACP in sugar-free chewing gum can remineralize enamel subsurface lesions [11].

CPP-ACP improves teeth enamel’s resilience to acids and may have an impact on adhesion by interfering with acid etching. The literature data on CPP-ACP impact on orthodontics, however, are limited. When teeth considered for orthodontic bracket bonding were pretreated with CPP-ACP, Keçik et al. discovered that the shear bond strength was significantly improved. In a previous experiment, Dunn hypothesized that orthodontic brackets attached to teeth with a composite material containing ACP failed at considerably lower stresses than brackets bonded to teeth with a typical resin-based composite orthodontic cement. The outcomes of other trials, unrelated to brackets, were likewise contradictory. According to the literature, if the enamel is treated with CPP-ACP, the shear bond strength of resin to enamel when employing a self-etching priming adhesive (Clearfil SE Bond) as opposed to an all-etch adhesive (Single Bond) may be impacted. The use of a CPP-ACP-containing remineralizing paste (Tooth Mousse, GC Corp, Tokyo, Japan) did not appear to have an impact on the microshear bond strength to enamel for either Single Bond or Clearfil SE Bond, according to the study by Adebayo et al. The contradictory results could be explained by technique sensitivity [11].

Moreover, in their study, Scribante et al. observed that the pretreatment of the enamel with erythritol lowered the failure rates of brackets [12].

Demineralization, in essence a chemical process, becomes clinically manifest with the appearance of chalky white spots and is a phenomenon that accompanies orthodontic treatment, being able to represent a complication of it when it is not controlled. One of the main causes of enamel demineralization during orthodontic treatment is represented by the rough surface that remains around the attachments after applying the adhesive, which leads to the accumulation of bacterial plaque, a specific favoring factor. Demineralization can be controlled or prevented by using products containing fluoride, amorphous calcium phosphate, hydroxyapatite, tetra calcium phosphate, etc. Saliva constantly supplies the surface of the tooth with fluoride and other remineralizing components, preventing tooth demineralization and stimulating remineralization [7,13].

The brackets placements increases dental plaque retention and favor bacterial invasion around the brackets as a result of the change in the microbial environment in that area, which can lead to enamel demineralization, a process that begins 4 weeks after the adhesive application of the brackets [13].

The most effective method to prevent the occurrence of caries is through the topical application of fluoride; thus, the calcium in hydroxyapatite is replaced by fluorine and fluorapatite is formed, which has a much lower solubility than the original hydroxyapatite. Fluorapatite has two main advantages over hydroxyapatite. First, fluoride acts as a catalyst helping to remineralize the enamel with phosphate ions dissolved in saliva. Second, replacing the hydroxide group with fluorine reduces the dissolution of hydroxyapatite by lactic acid [3,6].

Conventional orthodontic bracket bonding techniques involve enamel pre-treatment with 37% PA gel to ensure secure attachment of the bracket to the enamel surface. Etching with 37% PA removes the outermost enamel layer and creates surface irregularities and micropores to enhance adhesive material infiltration, resulting in optimal micromechanical retention with the enamel surface. Fixed orthodontic appliances, through the specific attachments, increase the individual risk of caries and predispose the enamel to demineralization, which is why pastes with remineralization properties are frequently recommended. 

Another factor that influences the SBS is represented by light curing time. In their study, the authors recommend the usage of six seconds of LED light curing at 2500 YmW/cm^2^ during the bracket bonding process to obtain a satisfactory SBS [14].

There are no clear guidelines in the literature regarding bond strength limits, but the material should permit good adhesion to sustain masticatory forces (5–10 MPa minimum shear bond strength) and not exert excessively strong adhesion forces that can lead to enamel loss when debonding (40–50 MPa) [15].

The aim of the study is to evaluate the influence of remineralizing agents on the SBS. The null hypothesis suggested that remineralizing agents did not exert direct effect on orthodontic brackets’ SBS.

## 2. Materials and Methods

### 2.1. Ethical Approval

This in vitro study was approved by the Ethics Committee of the University of Medicine and Pharmacy “I. Hatieganu” Cluj-Napoca (No. 157/10.06.2022).

### 2.2. Preparation of Samples: Inclusion/Exclusion Criteria

A total of 40 freshly extracted teeth for orthodontic purposes (premolars and thirds molars) were collected. For the selection, the inclusion criteria were as follows: absence of caries, absence of volume, shape and structure anomalies, absence of restorations, and absence of traumatic lesions. Human teeth were considered for this study to achieve the most accurate results.

The enamel surface of each tooth was scaled and polished with a rubber polishing cup and pumice using a low-speed handpiece for 10 s, then stored in deionized water at room temperature (27 °C) for 48 h. 

Brackets (Evolve Low Profile, 0.22, Roth prescription, DB Orthodontics Ltd., Silsden, England) were placed on vestibular enamel in accordance with manufacturer recommendations: acid etching with orthophosphoric acid 37% for 30 s (Meta biomed, Colman, PA, USA); rinsing and drying with the water–air spray; applying the adhesive (Bond-It Light Cure Sealant^®^, DB Orthodontics, UK) with an applicator; drying and light curing for 10 s using O-Star Woodpecker cordless curing light (Guilin Woodpecker Medical Instrument Co., Ltd., Guilin, China); bonding material application (Bond-it Light cure adhesive^®^, DB Orthodontics, UK) on the bracket base; removing excess and checking the position; and carrying out photopolymerization 30 s from different angles. 

### 2.3. Demineralization–Remineralization Protocol

The teeth were randomly divided into four groups; groups I, II, and III were prepared for the demineralization–remineralization process. For demineralization, the teeth were immersed in 0.1% citric acid for 30 min, twice a day, for 20 consecutive days. The teeth in the control group (C) were immersed only in artificial saliva. 

After the demineralization process, remineralization products with subsequent composition were used (Table 1).

### 2.4. Preparation of Groups

The teeth were divided into 4 study groups, in accordance with a specific remineralization algorithm:-Group I: Elmex Sensitive professional^®^ toothpaste and GC MI Paste Plus^®^-Group II: Elmex Sensitive professional^®^ toothpaste and GC Tooth Mousse^®^-Group III: Elmex Sensitive professional^®^ toothpaste-Group C (control group): Elmex Sensitive professional^®^ toothpaste.

After each demineralization procedure (immersion in citric acid), teeth were brushed with Oral-B^®^ Braun electric toothbrush (Braun, Melsungen, Germany) using Oral-B^®^ 3D White brush head and toothpaste for one minute, on all surfaces of the dental crown. After toothbrushing, remineralizing pastes (MI Paste Plus^®^ and GC Tooth Mousse^®^, Leuven, Belgium) were applied with an applicator on each tooth and left to act for 3 min, after which the teeth were rinsed with distilled water.

In order to assess the SBS, at the end of the 20 days, the roots of the teeth were embedded in epoxy resins. After this process, the brackets were removed from the tooth surface and the shear strength was tested for each group. After the demineralization–remineralization procedures, teeth were kept in artificial saliva with a pH of 7.4, at 37 °C, until need. By recording the shear resistance, the adhesion of the brackets to the tooth surface was evaluated. 

SBS was assessed and recorded using the ASTM D638 compression test of a Lloyd LR5k Plus dual-column mechanical testing machine (Ametek/Lloyd Instruments, Meerbusch, Germany), with the maximum allowed capacity of 5 kN. The device is equipped with an electronic system for transmitting and measuring the compression force, choosing a force of 0.5 N that moves with 1 mm/minute. Each bracket was stressed vertically (with an occluso-gingival direction). The load required to detach the brackets was measured in newtons and then converted into megapascals (MPa). Data were processed using NexygenPlus software (2009, Lloyd Instruments, UK).

### 2.5. Statistical Analysis

The value of the adhesive strength of the brackets for each study group was represented by the average of 21 mechanical tests. Data were subjected to ANOVA tests and Tukey tests for post hoc comparisons between groups, and the significance level was set at *p* < 0.05, using Origin2019 Graphing & Analysis (Massachusetts, USA) software.

## 3. Results

Brackets adhesion to enamel surface varies between groups, in accordance with specific treatment (Table 2 and Table 3).

Prior to detachment, the mechanical process describes a curve until maximum resistance was reached and the bracket was lost (Figure 1). 

Group I and II (for which CPP-ACP agents were used as remineralization agents) express greater values for SBS, with statistically significant differences (*p* < 0.05 when compared with teeth exposed only to fluoride toothpaste) (Figure 1 and Figure 2).

Considerable differences were recorded between the control group (C) and groups I (Elmex Sensitive professional^®^ toothpaste and GC MI Paste Plus^®^) and II (Elmex Sensitive professional^®^ toothpaste and GC Tooth Mousse^®^) (*p* < 0.05) and between groups II (Elmex Sensitive professional^®^ toothpaste and GC Tooth Mousse^®^) and III (Elmex Sensitive professional^®^ toothpaste) (*p* < 0.05). Between the control group and group III (Elmex Sensitive professional^®^ toothpaste) and between groups I (Elmex Sensitive professional^®^ toothpaste and GC MI Paste Plus^®^) and II (Elmex Sensitive professional^®^ toothpaste and GC Tooth Mousse^®^), the differences were not significant (*p* < 0.05). The highest SBS of the brackets was identified for group II followed by group I, were fluoride toothpaste and CPP-ACP products were used.

## 4. Discussion

The results reject the null hypothesis that remineralizing agents exert a direct role on the SBS of orthodontic brackets.

In the literature, many techniques and materials have been investigated in order to test remineralizing agents’ effect on enamel structural and mechanical properties. Most studies explored fluoride-based varnishes or casein phosphopeptide amorphous calcium phosphate pastes, tested for individual or combined use. The influence on SBS of preventive measures of caries that include remineralizing agents is controversial [16,17,18,19].

In this study, the mean SBS for all groups fell within the Reynolds recommended range of optimal bracket adhesion equal to 5.9–7.8 MPa except for groups III and IV [20].

In our study, the adhesion test was performed on demineralized enamel and remineralized enamel. The results showed that shear bond strength values after remineralizing treatment were higher when compared to the control group or the group where only the fluoride containing toothpaste was used.

These results can be explained by the fact that groups I and II were treated similarly, with toothpaste and remineralizing pastes, while group III and the control group were treated only with toothpaste [21,22].

The development of caries during and after orthodontic treatment is a significant issue. Prevention of caries and orthodontic treatment are closely related, due to the high prevalence of caries and occlusal anomalies, which rank first and third in all oral diseases, respectively [23].

Reynolds indicates that the appropriate values for adhesion strength in orthodontic treatment are between 5.9 and 7.8 MPa. In the present study, the results for groups I (10,365 MPa) and II (14,205 MPa) treated with the pastes containing CPP-ACP and fluoride (MI Paste Plus) and CPP-ACP (Tooth Mousse), respectively, show higher values than those indicated [10, 16]. This result coincides with those produced by other researchers who evaluated the effects of CPP-ACP and fluoride, on shear strength SBS, applied to demineralized enamel before bracket attachment. They observed that the sheer force of teeth with demineralized and untreated enamel is lower; this fact may be due to the poor quality of the enamel surface. As in the present study, they found that the SBS of teeth pretreated with CPP-ACP is higher than the teeth that went through the demineralization process and were not treated. Contrary to the present study, in which the control group has the lowest values of adhesion resistance (4.1138 MPa), other studies obtained the highest values in the control group, without significant differences between it and the group treated with CPP-ACP [24]. 

Considering the results obtained and studies performed by other researchers, it can be assumed that the prophylactic use of fluoride alone leads to lower bracket retention. This may be since fluoride ions replace calcium ions in the surface layer of the enamel to form fluorapatites, which are more resistant to environmental influences, as well as phosphoric acid, which is used to etch the enamel. As a result, the depth of penetration of the bonding system into the enamel may decrease [25].

Such a potential negative effect of fluoride on SBS should not outweigh its positive effects on prophylaxis [26]. Enamel mineral loss occurs initially when pumicing and etching the enamel surface to induce demineralization and generate a microporous surface to provide micromechanical retention between the enamel and adhesive material. Further enamel demineralization and white spot formation can occur throughout the prolonged treatment duration (an average of 12–18 months) [27,28]. 

Individual and professional oral hygiene techniques both aim to remove the biofilm. The use of fluoride-based products has always been the gold standard for the remineralization of hard tissues, thanks to its ability to integrate into the enamel transforming the hydroxyapatite in fluorapatite [29,30]. However, recent research has moved towards the introduction of new remineralizing techniques alternative to fluoride and based on the integration of calcium and phosphates at the level of demineralized dental surfaces. The main benefits of these materials are attributed to their ability to localize at tooth surface and incorporate into supragingival plaque to provide bioavailable calcium (Ca) and phosphate (P) ions in particular and critical conditions where they are most needed [31,32]. 

The bond strength of the bracket can be affected by the changes In the mineral content of the tooth (especially after orthodontic attachments bonding) which can increase the porosity and permeability of the enamel, reducing its microhardness [33,34].

Moosavi et al. obtained results similar to ours when studying the effects of CPP-ACP-containing pastes on SBS. In their study, they analyzed the effect of an Er:YAG laser combined with the CPP-ACP pastes. However, after the combined application of MI paste and the laser in their investigation, the SBS values dramatically decreased. The results of the study demonstrate that applying MI Paste©, even while simultaneously using an Er:YAG laser, can increase dentin’s SBS. According to Kamozaki et al.’s findings, the application of self-etch adhesive to softened dentin surfaces exposed to laser radiation decreased the bond strength, while the application of MI Paste in combination with laser radiation had no effect on the bond strength. It appears that the destruction of organic dentin components and changes in the surface morphology of the tooth, followed by a decrease in calcium and phosphate in the dentinal substrate and changes in the composition of hydroxyapatite, may be the causes of the laser group’s reduced bond strength. Due to the high absorption spectra of Er:YAG in water and hydroxyapatite, it is thought that following Er:YAG laser radiation, the temperature in the irradiated surfaces abruptly increases. The absorbed energy in the surface layer causes the surface layer to degrade [35].

White spot lesions (WSLs), also known as enamel demineralization, are a common unintended side-effect following orthodontic treatment with fixed equipment. Up to 96% of orthodontic patients have the lesions, which are frequently seen on the labial surfaces of the maxillary incisors. The potential of WSL to improve following bracket removal is limited, affecting the treatment’s cosmetic outcome [36].

Professional applications of fluoride varnish are a cornerstone in the prevention of both primary and secondary caries, and their effectiveness has been proven in several systematic reviews. When compared to a placebo or no treatment, fluoride varnish treatments were prevented for 43% of permanent teeth and 37% of the primary dentition in a Cochrane Library update [36].

These lesions often develop after four weeks if no anticariogenic agent is used, which highlights their fast occurrence. High prevalence of enamel decalcification during fixed orthodontic treatment is partly attributed to the irregular bracket surface and presence of orthodontic wires, bands, and other attachments, which enhance plaque retention, complicate oral hygiene and limit the self-cleaning capacity of teeth with the salivary flow and movement of oral muscles. Consequently, the plaque pH drops due to the presence of fermentable carbohydrates, faster accumulation and maturation of plaque, and colonization of aciduric bacteria such as Streptococcus mutans and Lactobacilli. Elimination of microbial plaque after orthodontic treatment is not sufficient for treatment of WSLs because secondary lesions may develop 5–12 years after completion of orthodontic treatment. On the other hand, natural remineralization by the mineral ions present in the saliva only occurs in the superficial layer of WSLs [36].

The management of the postorthodontic white spot lesions consists of three methods. The first one, natural remineralization, can be accomplished using topical fluoride supplements and maintaining appropriate dental hygiene habits throughout the course of treatment, which can prevent or eliminate the occurrence of white spot lesions. The second method, the physical removal of discolored enamel by microabrasion with the use of 18% hydrochloric acid (HCl) and polishing with pumice powder, improves aesthetics. The microabrasion technique, which is frequently used to treat WSLs and induces surface abrasion, is quite invasive and results in tooth structure loss. It can remove up to 250 μm of enamel. The third method involves the usage of the CPP-ACP trypsin to digest the milk protein casein, which then forms casein phosphopeptide-stabilized amorphous calcium phosphate nanocomplexes, also known as Recaldent©, when combined with calcium and inorganic phosphate ions. In order to maintain a highly supersaturated solution, CPP-ACP stabilizes the calcium and phosphate ions, preventing their transition into crystalline phases. At a rate of 1.5–3.9 × 10^−8^ mol hydroxyapatite/m^2^, CPP stabilized calcium phosphate solutions can remineralize enamel subsurface lesions. The usage of fluoride dentifrice for the next three months after using CPP-ACP for three months aids in the regression of postorthodontic white spot lesions (WSLs) in terms of area, color, and durability, according to the literature. Applying 10% CPP-ACP for three months, coupled with regular oral hygiene practices and fluoridated toothpaste (1000 ppm), significantly reduced the severity of postorthodontic white spot lesions. After three months, there was a significant reduction in white spot lesions, and over the course of a year, achievement scores remained consistent [36].

## 5. Conclusions

Whitin the limitations of this in vitro study, the subsequent conclusions can be formulated:

Preserving the mineral content of the enamel requires supplementing oral hygiene with products with specific action.

The remineralizing properties of the CPP-ACP pastes increased the shear bond strength, an element that allows us to consider their use for preserving the morphological and structural characteristics of dental enamel and preventing dental caries during orthodontic treatment.

The use of fluoridated toothpaste alone can reduce the mechanical properties of the orthodontic adhesive, with a negative impact on brackets’ SBS and orthodontic mechanics. 

## Figures and Tables

**Figure 1 children-10-00268-f001:**
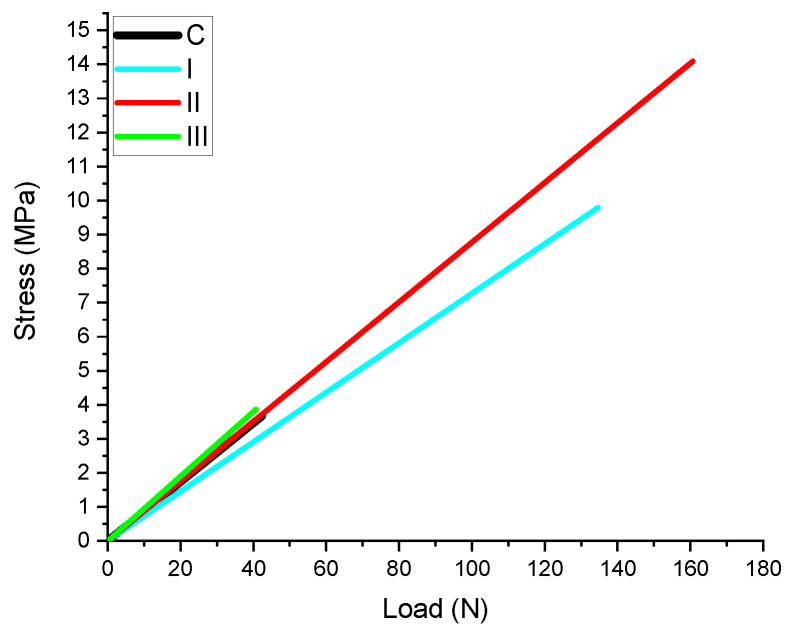
Deformation to breaking curve group. (N—newton, MPa—MegaPascal).

**Figure 2 children-10-00268-f002:**
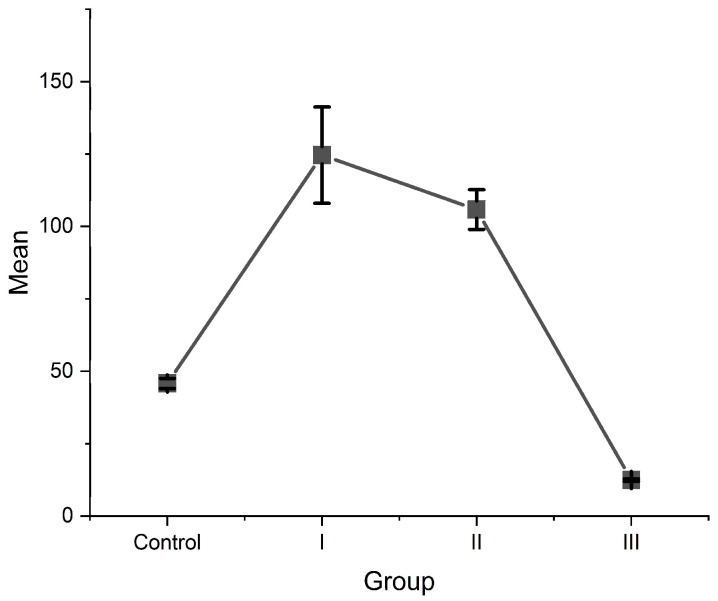
Shear bond strength—Tukey test.

**Table 1 children-10-00268-t001:** Composition of the remineralization agents.

Product	Ingredients	Effect
ELMEX^®^ Sensitive Proffesional Pro-argin toothpaste, Colgate-Palmolive, Gaba GmbH, Germany	Arginine, Calcium Carbonate, Aqua, Sorbitol, Bicarbonate, Sodium Lauryl Sulfate, Sodium Monofluorophosphate, Aroma, Sodium Silicate, Cellulose Gum, Sodium Bicarbonate, Titanium Dioxide, Potassium Acesulfame, Xanthan Gum, 1450 ppm fluoride	Dental sensitivity, for adults and children from 7 years; closes the nerve channels of sensitive teeth. With regular use, it creates a protective barrier that remains intact even after exposure to acids.
GC MI Paste Plus^®^, GC, Leuven, Belgium	pure water, glycerol, CPP-ACP, d-sorbitol, CMC-Na, propylene glycol, silicon dioxide, titanium dioxide, xylitol, phosphoric acid, sodium fluoride, flavoring, sodium saccharin, ethyl p-hydroxybenzoate, propyl p-hydroxybenzoate, butyl p-hydroxybenzoate	For remineralizing and inhibiting initial caries lesions; during and after orthodontic treatment, especially on white spots; for providing extra protection, especially against acid attacks
GC Tooth Mousse^®^, GC, Leuven, Blegium	Pure water, Glycerol, CPP-ACP, D-Sorbitol, Silicon Dioxide, CMC-Na, Propylene glycol, Titanium dioxide, Xylitol, Phosphoric acid, Guar gum, Zinc Oxide, Sodium Saccharin, Ethyl p-hydroxybenzoate, magnesium oxide, Butyl p-hydroxybenzoate, Propyl p-hydroxybenzoate.	Effective in protecting teeth from tooth decay and erosion, buffering dental plaque pH, remineralizing white spot lesions and reducing dentine hypersensitivity.

**Table 2 children-10-00268-t002:** Mechanical Test results.

Group	Shear Strength (*N*)	Maximum Resistance (*N*)	Shear Bond Strength (MPa)
I	141.24	142.52	10.36
II	112.67	161.93	14.20
III	12.80	44.84	4.256
C	47.51	47.51	4.11

**Table 3 children-10-00268-t003:** Shear bond strength-one way ANOVA analysis for evaluated groups.

Group	*p* Values
I-II	>0.05
C-I	<0.003
C-II	<0.001
C-III	>0.06
II-III	<0.003
II-III	>0.05

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
