# Peer review of "Effect of Remineralizing Agents on Shear Bond Strength of Orthodontic Brackets—In Vitro Study"

_children, 2023, doi:10.3390/children10020268_

Round 1
Reviewer 1 Report
The study design is adequate. But overall major revision is required. I think it will be ready for publication once these corrections are made:
GENERAL EVALUATION:
There are conceptual confusions in terms of the terminology used in the manuscript.
Writing the chosen terminological words or expressions in the same way everywhere in the manuscript will improve the understanding of the study. For example, while ‘shear bond strength’ was written in the study title, it was written as ‘adhesion’ later on, and then it was written as ‘shear stregth’ in different places in the manuscript. In terms of completeness and intelligibility, it would be appropriate to choose the goal words/expressions correctly and to write them in the same way everywhere.
Please decide whether you will evaluate the study as ‘adhesion resistance’ or ‘shear strength’. Update the title and manuscript accordingly.
You should use abbreviations after specifying them in parentheses in the first place you write.
General English grammar revision required (spelling errors).
TITLE:
page 1, lines 3; please improve the type of study; ‘An In-vitro Study’ or ‘An In Vitro Study’ instead of ‘in-vitro study’
Please put the ‘-‘ dash between ‘in’ and ‘vitro’ either everywhere or not to put anywhere.
ABSTRACT
page 1, lines 16; It would be more appropriate to use ‘investigate’ instead of ‘analyze’ and ‘agents’ instead of ‘materials’ everywhere
page 1, lines 17; ‘bracket’ instead of ‘attachment’. With the word ‘adhesion’, the situation should be handled as in the ‘General Evatuation’ section mentioned above, and should be reviewed and developed everywhere from the title of study to the conclusion
You should specify that type of bracket are tested
You must specify the statistical analyzes performed with names of statistical test and the p significance value considered
page 1, lines 28: This subheading ‘Results’ should be removed or previous subheadings should be added
You should specify the results in terms of statistical significance.
The conclusions of the study should also be included in abstract
page 1, lines 30, keywords: ‘bracket’ and ‘adhesive’ keywords should be added in my opinion
What does the number ‘3’ in the keyword ‘orthodontic treatment 3’ mean? If by mistake, must be removed
INTRODUCTION
page 2, lines 71-74: This paragraph should be rephrased. ‘placement of brackets’ instead of ‘application of brackets’. Please rephrase this paragraph as ‘increases retention areas for bacterial invasion’.
page 2, lines 90-93: The aim of the study should be rephrased in a more understandable way to eliminate terminological conceptual confusion.
Introduction section is too brief. At least 1 paragraph consisting of 6-7 lines should be added to the Introduction containing up-to-date information on the effect of remineralizing agents on shear bond strength. (Isn’t that your main topic?)
MATERIALS AND METHODS
The ‘Materials and Methods’ section should be written in sequential order as headings. (For example; Ethical Approval, Preparation of Samples, Storage Conditions, Remineralizing and Brackets Used, Preparation of Groups, Fatigue Testing, Statistical analysis)
Under each heading, the subject should be explained in a regularly.
Important considerations such as inclusion/exclusion criteria, sample size calculation, and bonding method chosen should be cited with current references.
RESULTS
The table design should conform to that specified in the template.
Abbreviations written in the all tables should be stated below the tables, separately.
For results with or without statistically significant differences, p values should be stated as p<0.05 or p>0.05 at the end of the sentence.
There should be a table with statistical significance by specifying the p values of the groups.
DISCUSSION
Discussion section is too brief too.
Paragraphs in which study results are compared and/or correlated with current articles (2016 and later) should be added.
Please add a paragraph with ‘limitations and strenghts of the study’
Please add a paragraph with ‘limitations and strenghts of the study’
REFERENCES
References should be prepared in accordance with the template.
In the references, it is seen that some journal names are written short and some are long. Please correct them all according to the template.
Due to the fact that you have made a study in which the current literature can be easily accessed due to its subject, at least half of the total number of references should consist of articles from the last 5 years (2017 and later), in my opinion. (please add/update in discussion)
Reviewer 2 Report
Dear Authors,
Attached please find the review report.
Good work.

Reviewer 3 Report
Extensive editing of English language and style required
Abstract: Please remove “. The null hypothesis tested was that the adhesive properties of the orthodontic adhesives are not affected by the use of tooth pastes or other products with a remineralization effect.”
Materials& Methods: Pleases add the references for the protocol of demineralization and remineralization process used in this study.
Materials& Methods: Pleases mention sample size calculation
Results: Table 1: Standard deviation and Minimum of the Shear bond strength should be added
Results: Pleases add a table for two by two comparison test of shear bond between the groups
Discussion: Please explain the chemical and physical reasons of this phenomena: “The presents results showed that shear bond strength values after remineralizing treatment were higher when compared to control group or the group where only the fluoride containing tooth paste was used.”
Round 2
Reviewer 1 Report
Effect of remineralizing agents on shear bond strength of orthodontic brackets- in-vitro study
Round 2
Reviewer report
- I thank the authors for efforts to improve their study.
- However, there are still places where revisions are not performed in the manuscript and the first report file needs to be reviewed again! It is seen that not all points highlighted in the first report were reviewed.
- The revised pdf file was formed in a very messy layout and very difficult to read and understand.
- It would be more understandable and readable if indicated the revisions made in the text and references with yellow coloured highlighting.
- Add ‘remineralizing agents’ to keywords
- There is still no table regarding the statistical p values of shear bond strengths.
- I do not think that the study has revised to the extent that it can be published in this journal.

Reviewer 2 Report
Dear Authors,
Thank you for providing the revised version of the manuscript. I have noticed that just few points were addressed and minimum modifications have been performed. Moreover, it was not provided a point-by-point response for each reviewer. Choose a better way to display the modifications on the manuscript, because as it was performed it is very confusing.
Each point of the three Reviewers should be addressed, otherwise the manuscript will be rejected.
Please, carefully follow the former Review Reports.
Good work.
Reviewer 3 Report
The manuscript has been improved and it could be published.
Author Response
Thank you your help!
Round 3
Reviewer 2 Report
Dear Authors,
Thank you for providing the revised version of your manuscript, now it is suitable for publication.
Thank you for the effort.